# Improved Applicability and Diagnostic Accuracy of the Novel Spleen-Dedicated Transient Elastography Device for High-Risk Esophageal Varices

**DOI:** 10.3390/diagnostics14070743

**Published:** 2024-03-30

**Authors:** Anita Madir, Mislav Barisic Jaman, Marko Milosevic, Petra Dinjar Kujundžić, Ivica Grgurevic

**Affiliations:** 1Department of Gastroenterology, Hepatology and Clinical Nutrition, University Hospital Dubrava, 10000 Zagreb, Croatia; anita.madir@mef.hr (A.M.); mbarisic2@kbd.hr (M.B.J.); pdinjar@kbd.hr (P.D.K.); 2School of Medicine, University of Zagreb, 10000 Zagreb, Croatia; 3Faculty of Pharmacy and Biochemistry, University of Zagreb, 10000 Zagreb, Croatia

**Keywords:** spleen stiffness measurement, transient elastography, high-risk esophageal varices, Baveno VII criteria

## Abstract

Spleen stiffness measurement (SSM) by transient elastography (TE) has been repeatedly demonstrated as the reliable way to rule out the presence of high-risk esophageal varices (HRV). We aimed to evaluate and compare novel vs. standard TE-SSM module performance in diagnosing HRV in patients with compensated advanced chronic liver disease (cACLD). This retrospective study included patients with cACLD; blood data, upper digestive endoscopy performed within 3 months of TE, SSM@50Hz and SSM@100Hz were collected. Overall, 112 patients with cACLD were analyzed (75.9% males, average age of 66, 43.7% alcohol-related chronic liver disease, 22.3% metabolic-associated steatotic liver disease, 6.2% viral hepatitis). Reliable SSM was possible in 80.3% and 93.8% of patients by using SSM@50Hz and SSM@100Hz probe, respectively. At the cut-off 41.8 kPa and 40.9 kPa (Youden), SSM@50Hz and SSM@100Hz had AUROCs of 0.746 and 0.752, respectively, for diagnosing HRV (*p* = 0.71). At the respective cut-offs, sensitivities for HRV were 92.9% and 100%, resulting in misclassification rates of 7.1% and 0% by using SSM@50Hz and SSM@100Hz. SSM reliably excludes HRV in cACLD patients, with measurements below 41 kPa potentially avoiding EGD in around 50% of cases, with minimal risk of HRV omission. SSM@100Hz demonstrated less measurement failures and no HRV misclassification.

## 1. Introduction

Portal hypertension (PH) is a common clinical syndrome characterized by a pathological increase in the portal vein pressure [1]. The most common cause of PH is chronic liver disease, i.e., cirrhosis [2]. PH plays a key role in the prognosis of chronic liver disease and is the main cause of morbidity and mortality in patients with liver cirrhosis [1,2]. The gold-standard diagnostic method for numerical quantification of the severity of PH is hepatic venous pressure gradient (HVPG) measurement, which is an invasive method accomplished through the catheterization on the hepatic veins under the fluoroscopic guidance [3]. Major complications of PH such as formation of esophageal varices (EV), accumulation of ascites, hepatic encephalopathy (HE), bleeding from EV and death develop in patients with clinically significant PH (CSPH), when HVPG reaches a threshold of ≥10 mmHg [4,5,6,7]. Invasiveness, high price and limited availability of HVPG have imposed the need to investigate alternative methods for assessing the severity of PH, such as elastographic techniques [8]. The gold standard for EV diagnosis is esophagogastroduodenoscopy (EGD), whereby EVs of any size and high-risk esophageal varices (HRV) are present in around 40–50% and 15–20% of patients with compensated advanced chronic liver disease (cACLD), respectively, making endoscopy unnecessary in a significant number of patients [9,10]. In the initial stage of development, PH is mostly influenced by the accumulation of fibrotic tissue within the liver; thus, liver stiffness measurement (LSM) has been demonstrated as the reliable diagnostic method to differentiate between patients according to the presence or absence of CSPH [11]. With further development and aggravation of PH, it becomes more influenced by hemodynamic factors, such as an increased blood flow in the splanchnic circulation, which cannot be captured by LSM. As a result, splenic congestion occurs, resulting in increased spleen stiffness [12]. Both LSM and spleen stiffness measurements (SSM) have thus become well established methods for noninvasive assessment of PH and its complications. SSM by transient elastography (TE) has been repeatedly demonstrated as the reliable way to rule out the presence of HRV [9,13]. A recent meta-analysis of 32 studies demonstrated overall very good diagnostic accuracy of SSM for diagnosing HRV (29.9% prevalence in the included studies), with an area under the receiver operating curve (AUROC) of 0.831, pooled sensitivity of 87%, specificity of 66%, positive predictive value (PPV) of 54% and negative predictive value of 88%. Based on these figures, it was calculated that 50.6% of patients might avoid endoscopy, with a risk of missing HRV of 8.4% following below-the-threshold SSM results (4.7% among the overall population of 2214 patients evaluated) [14]. Based on the accumulated evidence, the Baveno consensus suggested that SSM ≤ 40 kPa might be used to rule out HRV, in addition to the traditional criteria that include RV in patients in addition to the traditional criteria that LSM  ≤  20 kPa by TE and platelet count ≥ 150,000/mm^3^ in patients cACLD [15,16]. In 2018, a new spleen-dedicated module of TE operating at a fixed frequency of 100 Hz (featuring adjustable measurement depth and a stiffness range of 6–100 kPa and equipped with an ultrasound probe to localize the spleen) was introduced. This new method demonstrated lower rates of unreliable measurements and better accuracy for diagnosing HRV in several studies [17,18,19].

The aims of our study were to evaluate the overall diagnostic performance of SSM by TE in assessing the presence of EV and HRV in patients with cACLD, as well as to compare SSM@50Hz to the new SSM@100Hz in this indication.

## 2. Patients and Methods

This was a single-center retrospective cross-sectional study conducted in a tertiary care setting (University Hospital Dubrava, Zagreb, Croatia) over a period of 5 years (from January 2017 to December 2019 and from January 2022 to December 2023). The study protocol was conducted under the ethical guidelines of the Declaration of Helsinki (7th revision) and was approved by the Institutional Ethics Committee (Approval Number 20240307/01).

### 2.1. Study Population

Adult patients over 18 years of age presumed to have cACLD (based on LSM ≥ 15 kPa as measured by TE) or with histological evidence of advanced fibrosis/cirrhosis upon liver biopsy (who also had results from an EGD performed within 3 months from the time of TE) were considered candidates for this study. The studied population was collected in two periods: the first period was from January 2017 to December 2019, when they were prospectively enrolled in another study that assessed the diagnostic performance of LSM and SSM@50Hz by TE versus 2-dimensional shear-wave elastography (2DSWE) in diagnosing the severity of portal hypertension and presence of HRV (*n* = 51 patients) [20]; the second period was from January 2022 to December 2023, when an additional 61 patients were enrolled and assessed by the new SSM@100Hz method. In the period 2020–2022, the hospital served as a COVID facility and was not regularly available for non-COVID patients, which is why new patients could not be enrolled. All patients underwent a standardized clinical work-up to define the etiology of liver disease, including medication history, presence of metabolic syndrome components (central obesity, arterial hypertension, dyslipidemia and decreased glucose tolerance or presence of diabetes), and screening for viral and autoimmune liver disease, haemochromatosis and Wilson’s disease. All patients included in the first period underwent transjugular liver biopsy, whereas liver biopsy was performed upon the indication of the attending physician in the second period.

Patients’ data were retrospectively obtained from the Hospital Information System, including medical history, demographic data, laboratory data, results of EGD, liver histology and imaging. The protocol of the study is presented in Figure 1.

### 2.2. Elastography of the Liver and the Spleen

In the first group of patients, a Fibroscan Touch 502 (Echosens, Paris, France) machine was used for LSM and SSM. This device was equipped with 50 Hz probes in M and XL size and an automatic probe selection tool. LSM and SSM were performed on patients in the supine position after overnight fasting using either the right arm (for LSM) or left arm (for SSM) in maximal abduction by positioning the probe in the right (for LSM) or left (for SSM, usually in the left posterior axillary line) intercostal space [21]. Measurements were performed during a short apnea period in the neutral breathing position. The M probe was used initially in all patients for LSM and was then switched to the XL probe if suggested by the device. LSM and SSM were calculated as the median value of 10 measurements. Only those values with IQR/Median ≤ 30% were considered reliable.

In the second group of patients, FibroScan^®^ Expert 630 (Echosens, Paris, France) was used to assess LSM and SSM. The new device was equipped with liver probes in M and XL size, including a dedicated SSM module operating at 100 Hz available at M probe, with an ultrasound localization system incorporated in the device. The new Fibroscan^®^ software SmartExam (Echosens, Paris, France) was primally designed for risk stratification of patients with compensated advanced chronic liver disease and portal hypertension. For SSM@100Hz, we used the same conditions as for the before mentioned Fibroscan Touch 502 device, with the exception that only a dedicated M probe was used.

### 2.3. Esophagogastroduodenoscopy for the Assessment of Esophageal Varices

Esophagogastroduodenoscopy is considered the gold-standard method for diagnosing EV, and all patients included in this study had access to results of an EGD performed within 3 months from the time of LSM/SSM. EV were divided into three stages: 0—absence of EV; 1—small EV (straight veins in the distal esophagus slightly elevated above the mucosa, which collapse upon insufflation of air into the lumen of the esophagus); 2—large EV (tortuous submucosal veins that do not collapse upon air insufflation). Large EV or EV with cherry red spots (CRS) were considered as high-risk varices (HRV) [22,23].

## 3. Statistical Analysis

The Shapiro–Wilk test was utilized to assess the normality of the distribution of numerical variables. However, none of the numerical variables under analysis demonstrated a normal distribution. Instead, they were presented in terms of the median and interquartile range (IQR) and compared between groups using the Mann–Whitney U test. Categorical variables were represented as ratios and percentages and were compared between groups using the Χ^2^ test. Age was described using the median and range. Logistic regression was employed to examine the independent associations between various parameters. Only variables that were univariately significant were included in the model building process. Due to many statistically significant variables in univariate analysis, we used the backwards approach (inclusion criteria *p* < 0.05, exclusion criteria *p* > 0.2) during the model building in all multivariate analyses where all univariately associated variables, age and sex were considered [24]. The AUROC (area under the receiver operator characteristic curve), along with its corresponding 95% confidence interval (CI), was calculated to assess the ability of SSM in determining the presence of EV and HRV. The DeLong test was used to compare independent ROC curves. Optimal cut-offs for distinguishing between presence of EV and HRV were established using the Youden Index, which aims to maximize the sum of sensitivity (sens.) and specificity (spec.) [25]. Furthermore, we searched for SSM cut-offs that would yield > 90% specificity (to be used as a criterion for ruling in EV and HRV) and cut-offs that would have >90% sensitivity (to be used as a criterion for ruling out EV and HRV), if these goals were not achieved at the optimal cut-offs. For each cut-off, sensitivity, specificity, positive predictive value (PPV), negative predictive value (NPV), positive likelihood ratio (+LR) and negative likelihood radio (−LR) were calculated. Statistical significance was defined as *p* values < 0.05. All statistical analyses were carried out using MedCalc statistical software version 22.016 (MedCalc Software Ltd., Ostend, Belgium).

## 4. Results

### 4.1. Patients’ Characteristics

Our cohort consisted of 112 patients with cACLD (85 (75.9%) males, average age of 66, median BMI 27.6 kg/m^2^, 70 (67.3%) overweight, 65 (59.6%) with arterial hypertension, 30 (27.5%) with hyperlipidemia, 44 (40.4%) with type 2 diabetes). The most common etiologies of cACLD were alcohol (49 patients, 43.7%), metabolic-associated steatotic liver disease (MASLD, 25 patients, 22.3%) and combined MASLD with alcohol-related liver disease (MetALD, 13 patients, 11.6%). Reliable SSM was possible in 61/76 patients (80.3%) by using a Fibroscan Touch 502 device with SSM@50Hz probe, and an additional 10 patients were excluded as no advanced fibrosis/cirrhosis was confirmed at liver biopsy, thus leaving the final 51 (45.5%) patients eligible for the current study. Reliable SSM by using FibroScan^®^ Expert 630 with SSM@100Hz probe was possible in 61/65 (93.8%) patients, representing the remaining 54.5% of the patients eligible for the current study. The presence of EV and HRV was confirmed in 52 (46.4%) and 23 (20.7%) patients, respectively. The mean LSM and SSM measured by TE was 26.1 kPa and 42.5 kPa, respectively. Patients assessed with the old Fibroscan device were older, had arterial hypertension and dyslipidemia less frequently and had slightly higher serum sodium concentration, whereas there was no statistically significant difference in the gender, BMI, etiology of cACLD, MELD score, Child–Pugh score, Fib-4, APRI, ALBI, presence of EV, HRV and laboratory parameters apart from sodium between patients assessed with old and new Fibroscan software SmartExam (Echosens, Paris, France) (Table 1).

### 4.2. Diagnostic Performance of SSM for Assessing the Presence of Esophageal Varices in the Overall Cohort

In the overall cohort of 112 patients, SSM had an AUROC of 0.784 to detect the presence of any grade of EV at the cut-off 40.9 kPa (Youden), with 86.5% sensitivity, 71.7% specificity, 72.6% PPV, 86% NPV, 3.05 +LR and 0.19 −LR. Optimized SSM cut-offs (with >90% sensitivity and specificity) for ruling in and ruling out the presence of any grade of varices were, respectively, ≥66.5 kPa (91.7% specif.) and ≤30.6 kPa (92.3% sens.). Presence of EV would be missed in 13.5%, 73.1% and 7.7% patients and the number of spared endoscopies would be 62/112, 22/112 and 29/112 by using reported SSM cut-offs (40.9, ≥66.5 and ≤30.6 kPa, respectively) (Table 2).

At the cut-off 40.9 kPa (Youden), SSM had an AUROC of 0.751 to detect the presence of HRV, with 95.7% sensitivity, 54.5% specificity, 35.4% PPV, 98% NPV, 2.1 +LR and 0.08 −LR. SSM cut-off optimized for ruling in (>90% specificity) the presence of HRV was ≥74.5 kPa (specif. 93.2%). Presence of HRV would be missed in 4.3% and 73.9% patients, and the number of spared endoscopies would be 62/112 and 13/112 by using reported SSM cut-offs (40.9 and ≥74.5, respectively) (Table 2). The presence of HRV would be missed using the Baveno VII criteria (SSM ≤ 40 kPa) in 1 patient.

In the multivariate analysis using logistic regression, the presence of EV (OR 10.815, 95% CI 2.667–43.875, *p* = 0.001), lymphocytes (OR 0.308, 95% CI 0.104–0.912, *p* = 0.03), platelet count (OR 0.979, 95% CI 0.965–0.994, *p* = 0.005), INR (OR 0.031, 95% CI 0.001–0.735, *p* = 0.03), albumins (OR 0.561, 95% CI 0.332–0.949, *p* = 0.03) and LSM as measured by TE (OR 1.050, 95% CI 1.006–1.096, *p* = 0.02) were independently associated with SSM > 40.9 kPa indicating the presence of HRV (Table 3). Univariate analysis is shown in Appendix A.

### 4.3. Diagnostic Performance of SSM@50Hz for Assessing the Presence of Esophageal Varices

At the cut-off 41.8 kPa (Youden), SSM had an AUROC of 0.839 to detect the presence of any grade of EV patients as assessed by old Fibroscan software (SSM@50Hz, N = 51 patients), with 89.7% sensitivity, 77.3% specificity, 83.9% PPV, 85% NPV, 3.94 +LR and 0.13 −LR. Optimized SSM cut-offs (>90% sensitivity and specificity) for ruling in and ruling out the presence of any grade of EV were, respectively, ≥65.2 kPa (90.9% specif.) and ≤29.9 kPa (93.1% sens.). Presence of any grade of EV would be missed in 10.3%, 69% and 6.9% of patients, and the number of spared endoscopies would be 31/51, 12/51 and 14/51 by using reported SSM@50Hz cut-offs (>41.8, ≥65.2 and ≤29.9 kPa, respectively) (Table 4).

At the cut-off 41.8 kPa (Youden), SSM@50Hz had an AUROC of 0.746 to detect the presence of HRV, with 92.2% sensitivity, 51.4% specificity, 42% PPV, 95% NPV, 1.91 +LR and 0.14 −LR. SSM cut-off optimized for ruling in (>90% specificity) the presence of HRV was ≥69.1 kPa (94.6% specif.). Presence of HRV would be missed in 7.1% and 71.4% patients, and the number of spared endoscopies would be 31/51 and 7/51 by using reported SSM@50Hz cut-offs (41.8 and ≥69.1, respectively) (Table 4). The presence of HRV would be missed by using the Baveno VII criteria (SSM ≤ 40 kPa) in 1 patient.

### 4.4. Diagnostic Performance of SSM@100Hz for Assessing the Presence of Esophageal Varices

At the cut-off 40.9 kPa (Youden), SSM had an AUROC of 0.752 to detect the presence of any grade of EV in patients measured with new Fibroscan software (SSM@100Hz, N = 61 patients), with 82.6% sensitivity, 71.1% specificity, 63.8% PPV, 87.2% NPV, 2.85 +LR and 0.24 −LR. Optimized SSM cut-offs (>90% sensitivity and specificity) for ruling in and ruling out the presence of any grade of EV were, respectively, ≥71.1 kPa (92.1% specif.) and ≤30.6 kPa (91.3% sens.). Presence of EV would be missed in 17.4%, 69.6% and 8.7% of patients, and the number of spared endoscopies would be 30/61, 11/61 and 15/61 by using reported SSM cut-offs (>40.9, ≥71.1 and ≤30.6 kPa, respectively) (Table 5).

At the cut-off 40.9 kPa (Youden), SSM@100Hz had an AUROC of 0.782 to detect the presence of HRV, with 100% sensitivity, 58.8% specificity, 30% PPV, 100% NPV, 2.43 +LR and 0 −LR). SSM cut-off optimized for ruling in (>90% specificity) the presence of HRV was ≥75.9 kPa (94.1% specif.). Presence of HRV would be missed in 0% and 66.7% of patients, and the number of spared endoscopies would be 30/61 and 7/61 by using reported SSM cut-offs (40.9 and ≥75.9, respectively) (Table 5). The presence of HRV would be missed by using Baveno VII criteria (SSM ≤ 40 kPa) in 0 patients.

### 4.5. Comparison of Independent ROC Curves for Assessing the Presence of EV and HRV

There was no significant difference when comparing the AUROCs for the presence of EV or HRV by using SSM between the patients measured with @100Hz software and @50 Hz software (EV: 0.752 vs. 0.839, *p* = 0.34; HRV: 0.782 vs. 0.746, *p* = 0.71).

## 5. Discussion

In this retrospective study conducted on patients with cACLD, we determined that SSM is a reliable method for excluding HRV. With SSM values < 41 kPa, it was possible to avoid EGD in approximately 50% of patients, with a very low risk of missing HRV. Additionally, the new SSM@100Hz method showed significantly lower measurement failure and lower (0%) misclassification rate of HRV compared to the old SSM@50Hz software.

Measurement of SSM using TE has emerged as a potential alternative or adjunct to LSM measurements combined with platelet count, which have already been established as methods for excluding HRV and adopted by the Baveno Consensus [23]. Spleen stiffness reflects elevated pressure in the portal system, leading to splenic congestion, and therefore can be used to assess the severity of portal hypertension throughout its spectrum of development. This is advantageous compared to LSM, which only reflects the initial stage of portal hypertension, dependent on the amount of liver fibrosis. Colecchia and colleagues demonstrated high reliability of SSM in excluding EV (at SSM < 41.3 kPa) and HRV (at SSM < 46 kPa), with a negligible number of missed HRV cases (<5%) [9,13].

One of the main limitations of using the standard liver-dedicated TE probe was the overestimation of spleen stiffness, the existence of a ceiling threshold (set at 75 kPa), and a significant number of unreliable measurements (up to 20%). Therefore, since 2018, a new M probe with a frequency of 100 Hz has been introduced, along with the addition of an ultrasound probe for spleen imaging, enabling correct positioning of the FibroScan probe [26]. This has reduced the number of unsuccessful measurements to around 5% (2.9%–7.5%) according to the results of several studies, including our own [17,18,19,27].

The results of our study are compatible with similar studies that used the new FibroScan probe SSM@100Hz [17,18,19,27]. It is important to emphasize that our results are limited to the application of SSM exclusively in patients with cACLD, who represent the most appropriate population for non-invasive screening for HRV, given that the prevalence of EV and HRV in this population ranges from 40–50% and 15–20%, respectively, potentially rendering EGD unnecessary in a significant number of patients [9,28]. Similar to our results, Stefanescu and colleagues, in a multicenter prospective study, evaluated SSM@100Hz in a cohort of 260 patients with chronic liver diseases, predominantly of viral etiology, with the prevalence of EV and HRV being 63.5% and 26.5%, respectively [17]. In our study, the predominant etiology was alcoholic and steatotic liver disease, with a slightly lower prevalence of EV and HRV at 46.4% and 20.7%, respectively. Stefanescu et al. demonstrated that the new SSM@100Hz method enabled reliable SSM measurements in more participants compared to the old method (92.5% versus 76.0%, *p* < 0.001), which was also found in our study (93.8% vs. 80.3%, *p* = 0.02) [17]. In both studies, slightly better diagnostic performance of SSM@100Hz compared to SSM@50Hz for HRV was observed (AUROC 0.778 vs. 0.737, *p* = 0.105 in Stefanescu et al.’s study [17] and 0.782 vs. 0.746 in our study), with similar threshold values of SSM around 40–41 kPa, which is also very similar to the recommendations of the Baveno VII consensus.

In their study, Liu J. and colleagues found higher threshold values of SSM@100Hz for HRV in patients with non-viral etiology of cirrhosis (50.4 kPa) compared to viral etiology (43.4 kPa) [19]. Due to the small number of patients with viral hepatitis in our study, we were unable to perform a similar analysis. However, further research will be necessary to determine if there are significant differences in the diagnostic performance of SSM for HRV depending on the etiology of liver disease [19].

Our study is limited by a relatively small number of included participants, mixed etiology of liver disease, a retrospective design which inevitably leads to selection bias and the absence of a head-to-head comparison between the SSM@50Hz and SSM@100Hz methods. On the other hand, participants were carefully selected, and the study was exclusively limited to patients with cACLD, representing a group where the use of non-invasive tests for screening for HRV is most legitimate. The obtained results are compatible with the results of prospective studies that investigated the diagnostic performance of SSM for HRV, indicating a low risk of bias resulting from the limitations of our study.

In conclusion, our study demonstrates that SSM by using TE is a reliable method for non-invasive diagnosis of HRV in patients with cACLD. The new spleen-dedicated SSM@100Hz module enables reliable measurements in 95% of patients, and at SSM values < 41 kPa, EGD can be avoided in 50% of participants, with a 0% risk of missing HRV. Based on these characteristics, this new device seems to be superior to the old SSM@50Hz module of TE. Additional studies are needed to analyze the impact of the etiology of chronic liver disease on the performance of SSM in HRV diagnosis.

## Figures and Tables

**Figure 1 diagnostics-14-00743-f001:**
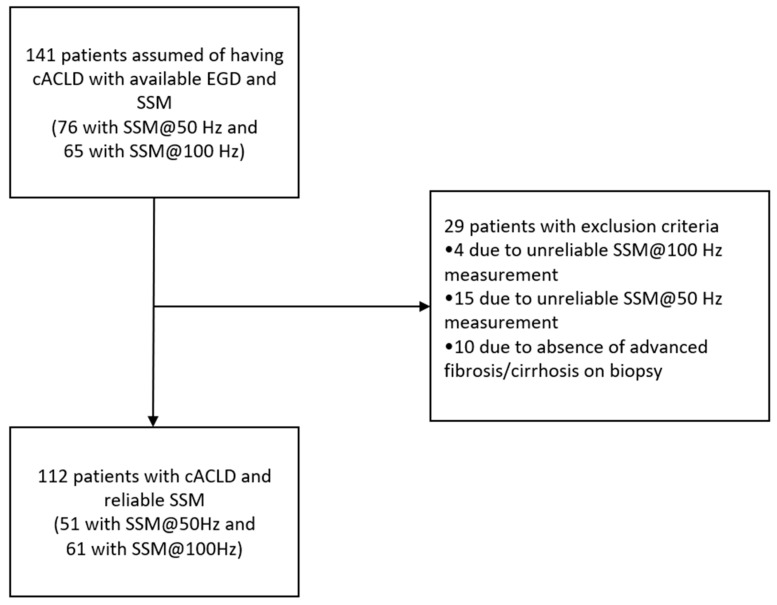
Flowchart of patient selection. Abbreviations: cACLD: compensated advanced chronic liver disease; EGD: esophagogastroduodenoscopy; SSM: spleen stiffness measurement.

**Table 1 diagnostics-14-00743-t001:** Characteristics of the patients included in this study. *p*-values were calculated using the Kruskal–Wallis test and the chi-squared test, comparing the values of patients who had SSM measured with old software and patients who had SSM measured with the new software. Significant *p*-values are in bold. Abbreviations: ALBI: albumin-bilirubin score; ALT: alanine aminotransferase; AST: aspartate aminotransferase; APRI: AST-to-platelet ratio index; BMI: body mass index; CAP: continuous attenuation parameter; dB/m: decibels per meter; ECOG: Eastern Cooperative Oncology Group; FIB-4: Fibrosis-4; GGT: gamma glutamyl transferase; g/L: grams per liter; INR: international normalized ratio; IQR: interquartile range; kPa: kilopascal; LSM: liver stiffness measurement; MMELD: model of end-stage liver disease; µmol/L: micromoles per liter; mmol/L: millimoles per liter; N: number; MASLD: metabolic dysfunction-associated steatotic liver disease; NAFLD: non-alcoholic fatty liver disease; PBC: primary biliary cholangitis; SSM: spleen stiffness measurement; U/L: units per liter.

	All Patients *n* = 112 (100%)	SSM@100 Hz *n* = 61 (54.5%)	SSM@50 Hz *n* = 51 (45.5%)	*p* Value
Age, years, median [IQR]	66 [57–71]	63 [56–70]	69 [62–71]	0.02
Sex				0.31
1-male	85 (75.9%)	44 (72.1%)	41 (80.4%)	
2-female	27 (24.1%)	17 (27.9%)	10 (19.6%)	
BMI, kg/m^2^, median [IQR]	27.6 [24.8–31.4]	28.4 [24.7–32.4]	26.9 [25.1–30.6]	0.23
ECOG				0.98
0	98 (89.9%)	55 (90.2%)	43 (89.6%)	
1	9 (8.3%)	5 (8.2%)	4 (8.3%)	
2	2 (1.8%)	1 (1.6%)	1 (2.1%)	
Obesity (BMI > 30 kg/m^2^)				0.27
Yes	70 (67.3%)	33 (62.3%)	27 (72.5%)	
No	34 (32.7%)	20 (37.7%)	14 (27.5%)	
Arterial hypertension				0.03
Yes	65 (59.6%)	42 (68.9%)	23 (47.9%)	
No	44 (40.4%)	19 (31.1%)	25 (52.1%)	
Hyperlipidemia				0.002
Yes	30 (27.5%)	24 (39.3%)	6 (12.5%)	
No	79 (72.5%)	37 (60.7%)	42 (87.5%)	
Diabetes type II				0.59
Yes	44 (40.4%)	26 (42.6%)	18 (37.5%)	
No	65 (59.6%)	35 (57.4%)	30 (62.5%)	
Charlson Comorbidity Index	5 [4–6]	4 [3–6]	5 [4–6]	0.39
Etiology				0.59
1-Alcohol	54 (48.2%)	32 (52.5%)	22 (43.1%)	
2-NAFLD	40 (35.7%)	22 (36.1%)	18 (35.3%)	
3-Viral hepatitis	7 (6.2%)	2 (3.3%)	5 (9.8%)	
4-PBC	2 (1.8%)	1 (1.6%)	1 (2.0%)	
5-Other	9 (8%)	4 (6.6%)	9 (8.0%)	
Esophageal varices				0.052
Yes	52 (46.4%)	23 (37.7%)	29 (56.9%)	
No	60 (53.6%)	38 (62.3%)	22 (43.1%)	
Grade of varices				0.16
No varices	60 (53.6%)	38 (62.3%)	22 (43.1%)	
Small varices	29 (25.9%)	14 (23.0%)	15 (29.4%)	
Large varices	23 (20.5%)	9 (14.8%)	14 (27.5%)	
High-risk esophageal varices				0.11
Yes	23 (20.7%)	9 (15.0%)	14 (27.5%)	
No	88 (79.3%)	51 (85.0%)	37 (72.5%)	
Leukocytes, G/L, median [IQR]	6.5 [4.7–7.8]	6.2 [4.4–7.6]	7.0 [4.9–7.8]	0.42
Neutrophils, G/L, median [IQR]	3.98 [2.67–5.12]	3.94 [2.39–4.98]	4.05 [2.85–5.50]	0.39
Lymphocytes, G/L, median [IQR]	1.50 [1.10–2.17]	1.50 [1.11–2.13]	1.50 [1.10–2.20]	0.94
Neutrophile-to-lymphocyte ratio, median [IQR]	2.42 [1.83–3.67]	2.42 [1.83–3.46]	2.40 [1.83–3.83]	0.69
Hemoglobin, g/L median [IQR]	139 [123–149]	139 [124–149]	139 [119–153]	0.92
Platelets, G/L, median [IQR]	143 [103–188]	137 [109–169]	156 [94–213]	0.27
INR, median [IQR]	1.10 [1.01–1.33]	1.10 [1.02–1.31]	1.08 [1.0–1.39]	0.99
Bilirubin, µmol/L, median [IQR]	20.2 [12.7–32.8]	22.3 [12.9–36.6]	19.0 [12.5–26.6]	0.22
Creatinine, µmol/L, median [IQR]	69 [57–78]	67 [56–76]	70 [63–83]	0.16
Sodium, U/L, median [IQR]	138 [137–140]	138 [136–139]	139 [138–140]	0.02
AST, U/L, median [IQR]	48 [33–75]	49 [32–79]	48 [35–69]	0.69
ALT, U/L, median [IQR]	37 [25–58]	33 [22–58]	40 [27–66]	0.22
GGT, U/L, median [IQR]	108 [61–223]	107 [59–273]	123 [65–215]	0.92
Albumins, g/L, median [IQR]	39 [35–44]	40 [36–43]	39 [34–44]	0.58
Child–Pugh score, points, median [IQR]	5 [5–6]	5 [5–6]	5 [5–6]	0.76
MELD, points, median [IQR]	10 [7–12]	11 [8–13]	10 [7–11]	0.11
FIB-4, points, median [IQR]	3.32 [2.34–5.74]	3.69 [2.61–6.14]	2.85 [2.13–5.27]	0.06
APRI, points, median [IQR]	0.96 [0.62–1.63]	1.03 [0.68–1.71]	0.87 [0.56–1.53]	0.12
ALBI, points, median [IQR]	−2.48 [−2.99–−2.08]	−2.48 [−2.98–−2.11]	−2.49 [−2.99–−1.95]	0.89
Spleen length, mm, median [IQR]	135 [115–150]	139 [122–150]	134 [114–144]	0.41
LSM, kPa, median [IQR]	26.1 [14.9–40.4]	26.5 [16.9–44.7]	25.1 [13.9–36.0]	0.16
CAP, dB/m, median [IQR]	270 [213–306]	269 [223–298]	276 [210–323]	0.63
SSM, kPa, median [IQR]	42.5 [30.6–59.3]	40.9 [30.7–58.8]	43.5 [29.9–61.0]	0.92
Reliable measurement (*n* = 141)				0.02
Yes	122 (86.5%)	61 (93.8%)	61 (80.3%)	
No	19 (13.5%)	4 (6.2%)	15 (19.7%)	

**Table 2 diagnostics-14-00743-t002:** Diagnostic performance of SSM for detecting the presence of any grade of esophageal varices and high-risk esophageal varices in all patients (*n* = 112). Abbreviations: AUROC: area under the receiver operator characteristics; EV: esophageal varices; HRV: high-risk esophageal varices; kPa: kilopascals; −LR: negative likelihood ratio; NPV: negative predictive value; +LR: positive likelihood radio; PPV: positive predictive value; SSM: spleen stiffness measurement.

EV Grade	Cut-Off	AUROC	SSM Cut-Off, Stiffness (kPa)	Sensitivity, %	Specificity, %	+LR	−LR	PPV, %	NPV, %	Missed Cases	Spared Endoscopy
Any grade of EV	Youden	0.784 (95%CI: 0.696–0.856)	40.9	86.5	71.7	3.05	0.19	72.6	86	7/52 (13.5%)	62/112 (55.3%)
	Rule in		≥66.5	25	91.7	3.00	0.82	72.3	58.5	38/52 (73.1%)	22/112 (19.6%)
	Rule out		≤30.6	92.3	41.7	1.58	0.18	57.8	86.2	4/52 (7.7%)	29/112 (25.9%)
HRV	Youden	0.751 (95%CI: 0.660–0.828)	40.9	95.7	54.5	2.1	0.08	35.4	98	1/23 (4.3%)	62/112 (55.3%)
	Rule in		≥74.5	26.1	93.2	3.83	0.79	50	82.8	17/23 (73.9%)	13/112 (11.6%)
	Rule out		--- ^a^	--- ^a^	--- ^a^	--- ^a^	--- ^a^	--- ^a^	--- ^a^	--- ^a^	--- ^a^

^a^ Sensitivity > 90% already at the optimal cut-off.

**Table 3 diagnostics-14-00743-t003:** Multivariate logistic regression model assessing independent predictors associated with SSM > 40.9 kPa (as the indicator of presence of critical varices) in our cohort of patients (*n* = 112). Abbreviations: ALBI: albumin-bilirubin score; CI: confidence interval; INR: international normalized ratio; IQR: interquartile ratio; LSM: liver stiffness measurement; SSM: spleen stiffness measurement.

Dependent Variable: SSM > 40.9 kPa	Odds Ratio (Multivariate)	95% CI	*p* Value for Multivariate Analysis
**Male sex**	2.906	0.638–13.232	**0.17**
**Esophageal varices**	10.815	2.667–43.875	**0.001**
**Lymphocytes, G/L, median [IQR]**	0.308	0.104–0.912	**0.03**
**Platelets, G/L, median [IQR]**	0.979	0.965–0.994	**0.005**
**INR, median [IQR]**	0.031	0.001–0.735	**0.03**
**Albumins, g/L, median [IQR]**	0.561	0.332–0.949	**0.03**
**ALBI, points, median [IQR]**	0.005	0.000–1.132	**0.06**
**LSM, kPa, median [IQR]**	1.050	1.006–1.096	**0.02**

**Table 4 diagnostics-14-00743-t004:** Diagnostic performance of SSM for detecting the presence of esophageal varices and critical esophageal varices using old Fibroscan software (*n* = 51). Abbreviations: AUROC: area under the receiver operator characteristics; EV: esophageal varices; HRV: high-risk esophageal varices; kPa: kilopascals; −LR: negative likelihood ratio; NPV: negative predictive value; +LR: positive likelihood radio; PPV: positive predictive value; SSM: spleen stiffness measurement.

EV Grade	Cut-Off	AUROC	SSM Cut-Off, Stiffness (kPa)	Sensitivity, %	Specificity, %	+LR	−LR	PPV, %	NPV, %	Missed Cases	Spared Endoscopy
Any grade of EV	Youden	0.839 (95%CI: 0.709–0.927)	41.8	89.7	77.3	3.94	0.13	83.9	85	3/29 (10.3%)	31/51 (60.8%)
	Rule in		≥65.2	31	90.9	3.41	0.76	81.8	49.9	20/29 (69%)	12/51 (23.5%)
	Rule out		≤29.9	93.1	54.5	2.05	0.13	73	85.7	2/29 (6.9%)	14/51 (27.4%)
HRV	Youden	0.746 (95%CI: 0.605–0.858)	41.8	92.9	51.4	1.91	0.14	42	95	1/14 (7.1%)	31/51 (60.8%)
	Rule in		≥69.1	28.6	94.6	5.29	0.76	66.8	77.7	10/14 (71.4%)	7/51 (13.7%)
	Rule out		--- ^a^	--- ^a^	--- ^a^	--- ^a^	--- ^a^	--- ^a^	--- ^a^	--- ^a^	--- ^a^

^a^ Sensitivity > 90% already at the optimal cut-off.

**Table 5 diagnostics-14-00743-t005:** Diagnostic performance of SSM for detecting the presence of esophageal varices and critical esophageal varices using new Fibroscan software (*n* = 61). Abbreviations: AUROC: area under the receiver operator characteristics; EV: esophageal varices; HRV: high-risk esophageal varices; kPa: kilopascals; −LR: negative likelihood ratio; NPV: negative predictive value; +LR: positive likelihood radio; PPV: positive predictive value; SSM: spleen stiffness measurement.

EV Grade	Cut-Off	AUROC	SSM Cut-Off, (kPa)	Sensitivity, %	Specificity, %	+LR	−LR	PPV, %	NPV, %	Missed Cases	Spared Endoscopy
Any grade of EV	Youden	0.752 (95%CI: 0.625–0.854)	40.9	82.6	71.1	2.85	0.24	63.8	87.2	4/23 (17.4%)	30/61 (49.2%)
	Rule in		≥71.1	30.4	92.1	3.86	0.76	69.9	68.6	16/23 (69.6%)	11/61 (18%)
	Rule out		≤30.6	91.3	34.2	1.39	0.25	45.6	86.7	2/23 (8.7%)	15/61 (24.6%)
HRV	Youden	0.782 (95%CI: 0.657–0.878)	40.9	100	58.8	2.43	0	30	100	0/9 (0%)	30/61 (49.2%)
	Rule in		≥75.9	33.3	94.1	5.67	0.71	49.9	88.9	6/9 (66.7%)	7/61 (11.5%)
	Rule out		--- ^a^	--- ^a^	--- ^a^	--- ^a^	--- ^a^	--- ^a^	--- ^a^	--- ^a^	--- ^a^

^a^ Sensitivity > 90% already at the optimal cut-off.

## Data Availability

The data presented in this study are available upon request from the corresponding author.

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
