# Peer review of "Improved Applicability and Diagnostic Accuracy of the Novel Spleen-Dedicated Transient Elastography Device for High-Risk Esophageal Varices"

_diagnostics, 2024, doi:10.3390/diagnostics14070743_

Round 1

Reviewer 1 Report

Comments and Suggestions for Authors

The manuscript reports on a study showing that spleen stiffness measurement by using transient elastography is a reliable method for diagnosis of high-risk esophageal varices in patients with compensated advanced chronic liver disease. Few points and typos to address and correct for in the minor revision of this manuscript:

·        Among the three acronyms catching the eyes in Figure 1, only SSM is addressed in its caption. Is there a reason for that? If not, please either address cACLD and EGD too or leave SSM also out as it is already covered in the Abstract and Introduction both.

·        It seems that in the caption of Table 1, all acronyms are addressed except NAFLD.

·        In captions of Tables 2, 4, and 5, “-PV” is typo and should be replaced with “-LR”.

·        The sentence ending at line 291 lacks the period.

·        In line 290, a blank space needed between 100 and Hz.

·        Rearrange the sentence starting in line 312 as “In their study, Liu J. and colleagues found …”

Considering the above, I appreciate the authors’ well-written manuscript and recommend it for publication in Diagnostics.

Author Response

DIAGNOSTICS

Special Issue “Ultrasound in Liver and Biliary Diseases”

Guest editors: Dr. Tudor Voicu Moga and Prof. dr. Alina Popescu

COVER LETTER

Dear Editors and Referees,

Thank you for reviewing our research article entitled “Improved applicability and diagnostic accuracy of the novel spleen-dedicated transient elastography device for high-risk esophageal varices” for publication in Biomedicines Special Issue " Ultrasound in Liver and Biliary Diseases”.

written by

Anita Madir, Mislav Barisic Jaman, Marko Milosevic, Petra Dinjar Kujundžić, and Ivica Grgurevic.

In accordance with your remarks, we are sending our corrected scientific article of improved quality. We believe that the corrected scientific article will be finally adequate for publication in Diagnostics Special Issue "Ultrasound in Liver and Biliary Diseases". All changes are highlighted.

  1. In accordance with the instructions of the first and second reviewer, we increased the quality of Figure 1, which was inserted in the scientific article in a better resolution. The image caption is supplemented with explanations of all abbreviations.
  2. NAFLD acronym was addressed in the caption of Table 1.
  3. A period has been inserted at the ending of sentence in line 291.
  4. A blank space was inserted between 100 and Hz in line 290.
  5. Rearrange The sentence starting in line 312 was rearranged as “In their study, Liu J. and colleagues found …”.

This paper has not been previously published in any language, nor is currently under consideration elsewhere for publication. Each author has approved the final draft submitted.

On behalf of co-authors

Yours sincerely,

Prof. Ivica Grgurevic, MD PhD, FEBG, FRCP

University of Zagreb School of Medicine

Faculty of Pharmacy and Biochemistry

Department of Gastroenterology, Hepatology and Clinical Nutrition

University Hospital Dubrava

Av. Gojka Suska 6

Zagreb 10000, Croatia

Reviewer 2 Report

Comments and Suggestions for Authors

Dear Authors this is a study about non invasive evaluation of patients suffering of compensated liver cirrhosis, well studied, complete of several variables under statistical evaluation. The limitations of the study are well underlined. There are some typo errors, above all spaces, and I think that the quality of figure 1 should be enhanced.  

Author Response

(The authors gave the same response as above.)
